# *FOXA3* Polymorphisms Are Associated with Metabolic Parameters in Individuals with Subclinical Atherosclerosis and Healthy Controls—The GEA Mexican Study

**DOI:** 10.3390/biom12050601

**Published:** 2022-04-19

**Authors:** Gilberto Vargas-Alarcón, José Manuel Fragoso, Julian Ramírez-Bello, Rosalinda Posadas-Sánchez

**Affiliations:** 1Department of Molecular Biology and Research Direction, Instituto Nacional de Cardiología Ignacio Chávez, Mexico City 14080, Mexico; gvargas63@yahoo.com (G.V.-A.); mfragoso1275@yahoo.com.mx (J.M.F.); 2Department of Endocrinology, Instituto Nacional de Cardiología Ignacio Chávez, Mexico City 14080, Mexico; dr.julian.ramirez.hjm@gmail.com

**Keywords:** atherosclerosis, cardiometabolic parameters, forkhead box, polymorphisms, subclinical atherosclerosis

## Abstract

*FOXA3* is a transcription factor involved in the macrophage cholesterol efflux and macrophage reverse cholesterol transport reducing the atherosclerotic lesions. Thus, the present study aimed to establish if the *FOXA3* polymorphisms are associated with subclinical atherosclerosis (SA) and cardiometabolic parameters. Two *FOXA3* polymorphisms (rs10410870 and rs10412574) were determined in 386 individuals with SA and 1070 controls. No association with SA was observed. The rs10410870 polymorphism was associated with a low risk of having total cholesterol >200 mg/dL, non-HDL-cholesterol > 160 mg/dL, and a high risk of having LDL pattern B and insulin resistance adipose tissue in individuals with SA, and with a high risk of having interleukin 10 <*p*25 and magnesium deficiency in controls. The rs10412574 polymorphism was associated with a low risk of insulin resistance of the adipose tissue and a high risk of aspartate aminotransferase >*p*75 in individuals with SA, and with a low risk of LDL pattern B and a high risk of a magnesium deficiency in controls. Independent analysis in 846 individuals showed that the rs10410870 polymorphism was associated with a high risk of aortic valve calcification. In summary, *FOXA3* polymorphisms were not associated with SA; however, they were associated with cardiometabolic parameters in individuals with and without SA.

## 1. Introduction

Coronary artery disease, the final consequence of the atherosclerotic process, is the leading cause of death worldwide [1]. Atherosclerosis includes three main processes, one the oxidative, in which the LDL particles that penetrate the subendothelial space are oxidized by the action of reactive oxygen species; the second the inflammatory process, which begins when the particles of oxidized LDLs are phagocytosed by resident macrophages at the injury site. During this process, many pro-inflammatory cytokines and chemokines are produced, which perpetuates the damage and leads to the formation of foam cells. Finally, these foam cells become covered with fibrin, forming atherosclerosis plaque, which ruptures, generating a thrombus that will occlude the artery [2]. Thus, macrophages have an essential role in the development and progress of atherosclerosis modulating foam cell formation and the inflammatory response. It has been suggested that macrophages’ reverse cholesterol transport is a protective mechanism for atherosclerosis due to this mechanism preventing macrophages from accumulating excessive cholesterol and, in consequence, diminishing the formation of foam cells [3]. Recently, Li et al. reported in an animal model that the overexpression of *FOXA3* in the liver increases the macrophage cholesterol efflux, and macrophage reverse cholesterol transport reducing the atherosclerotic lesions [4]. FOXA3 belongs to the forkhead box (FOX) proteins family: proteins that have been shown to transcriptionally control early development, organogenesis, and metabolism in mice [5]. FOXA3 promotes adipocyte differentiation and has been involved in developing insulin resistance and obesity related with age [6,7]. FOXA3 binds to regions of DNA located in the promoters of genes that code for tyrosine aminotransferase, phosphoenolpyruvate carboxykinase, and insulin-like growth factor-binding protein 1 [8,9,10], which function as insulin and glucocorticoid-response elements. The gene that encodes FOXA3 is located on chromosome 19 and is polymorphic. In 2015, Adler-Wailes et al. studied the possible association of variants in the *FOXA3* gene with metabolic parameters in a group of 392 lean and obese children, adolescents, and young adults [11]. They reported an association of the rs28666870 polymorphism with greater total lean body mass, increased BMI, and appendicular lean mass. Since then, no association studies of *FOXA3* polymorphisms with atherosclerosis or metabolic parameters have been conducted. Given the role that FOXA3 plays in macrophage cholesterol efflux and its possible association with insulin resistance, obesity, and atherosclerosis, we consider it interesting to carry out studies on the association of polymorphisms of this gene with cardiovascular diseases and metabolic parameters. Thus, the present study aimed to evaluate the association of two *FOXA3* polymorphisms with subclinical atherosclerosis (SA) and cardiometabolic parameters in a cohort well-characterized from the clinical, tomographic, and biochemical points of view.

## 2. Materials and Methods

### 2.1. Subjects

The study included 1456 healthy, asymptomatic individuals without a family history of premature coronary artery disease (pCAD) belonging to the Genetics of Atherosclerotic Disease (GEA) Mexican cohort. All individuals were recruited from the blood bank donors and written media invitations at social service centers. Tomography of the chest and abdomen was performed in these individuals using a 64-channel multidetector helical computed tomography system (Somatom Cardiac Sensation, 64, Forchheim, Germany). Information about total abdominal fat (TAF), subcutaneous abdominal fat (SAF), visceral abdominal fat (VAF) [12], liver and spleen attenuation [13], coronary artery calcium (CAC), and aortic valve calcification (AVC) was obtained by the tomography. The CAC and AVC were defined using the Agatston method [14]. After the computed tomography, 386 individuals were classified as individuals with SA (those individuals with CAC score >0) and 1070 as healthy controls (individuals with CAC score = 0). Excluded criteria were congenital heart failure, liver, renal, thyroid, and oncological disease, and premature CAD. Clinical, demographic, biochemical, and anthropometric parameters and cardiovascular risk factors were evaluated as previously described [15,16,17].

### 2.2. Genetic Analysis

Two *FOXA3* polymorphisms (rs10410870 and rs10412574) were determined using 5’ exonuclease TaqMan assays on an ABI Prism 7900HT Fast Real-Time PCR system (Applied Biosystems, Foster City, CA, USA). To corroborate the adequate assignment of the genotypes in the TaqMan assays, 10% of samples were randomly selected and repeated. These samples were 100% concordant in two independent assays.

### 2.3. Statistical Analysis

Data are expressed as frequencies, median (interquartile range), or mean ± standard deviation, as appropriate. Either Mann–Whitney U or Student’s *t*-test was used for continuous variable comparisons, while the Chi-Squared test was employed for categorical variable comparisons. Alleles and genotype frequencies were determined by direct counting. The Chi-Square test determined Hardy–Weinberg’s equilibrium. The association of the polymorphisms with SA and cardiometabolic parameters in SA and healthy controls were evaluated using logistic regression analysis under different inheritance models (additive, codominant 1, codominant 2, dominant, heterozygote, and recessive). The different models were adjusted for confounding variables as appropriate. Haploview version 4.1 (https://www.broadinstitute.org/haploview/haploview (accessed on 30 March 2022)) (Broad Institute of Massachusetts Institute of Technology and Harvard University, Cambridge, MA, USA) was used to establish linkage disequilibrium (LD, D’) and construction of haplotypes. The statistical power to detect association of the polymorphisms with SA was determined using the OpenEpi software [http://www.openepi.com/Power/PowerCC.htm (accessed on 30 March 2022)].

### 2.4. Functional Analysis

The Human-Transcriptome Database for Alternative Splicing (http://h-invitational.jp/, accessed on 20 October 2021), SNP Function Prediction (https://snpinfo.niehs.nih.gov, accessed on 20 October 2021), ESE finder (http://rulai.cshl.edu/cgi-bin/tools/ESE3/esefinder.cgi, accessed on 20 October 2021), Splice Port: An Interactive Splice Site Analysis Tool (http://spliceport.cbcb.umd.edu/SplicingAnalyser.html, accessed on 20 October 2021), SNPs3D (http://www.snps3d.org/, accessed on 20 October 2021) and HSF (http://www.umd.be/HSF/, accessed on 20 October 2021) bioinformatics tools were used to define the possible functional effect of the FOXA3 polymorphisms. These tools showed that both polymorphisms have possible functional effects. The rs10410870 produces a binding site for the CEBP transcription factor, whereas the rs10412574 produces binding sites for HMGIY, HNF4-alpha, SFI, and SRF transcription factors.

## 3. Results

### 3.1. Demographic, Clinical, and Biochemical Characteristics

Demographic, clinical, biochemical, and life style characteristics in the studied groups are shown in Table 1. Individuals with SA showed high levels of total cholesterol (*p* = 0.008), LDL-C (*p* < 0.001), non-HDL-C (*p* < 0.001), non-HDL-C > 160 mg/dL (*p* < 0.001), and low levels of HDL-C (*p* = 0.015) and magnesium (*p* = 0.002) when compared to healthy controls. In the same way, the percentage of individuals with total cholesterol >200 mg/dL (*p* < 0.001), non-HDL-C > 160 mg/dL (*p* < 0.001), insulin resistance of adipose tissue (*p* = 0.001), magnesium deficiency (*p* = 0.04), and aortic valve calcification (*p* < 0.001) was higher in the individuals with SA than healthy controls.

### 3.2. Association of the FOXA3 Polymorphisms with SA, Metabolic Parameters, and Cardiovascular Risk Factors

Genotype frequencies did not deviate from the Hardy–Weinberg equilibrium in any case (HWE, *p* > 0.05). The statistical power to detect an association with SA considering an unmatched case-control study was 0.80. The *FOXA3* polymorphisms were not associated with SA; however, both polymorphisms were associated with some cardiometabolic parameters in SA individuals and healthy controls. In SA individuals, rs10410870 was associated with a low risk of having total cholesterol >200 mg/dL (OR = 0.633 (0.421–0.954) *p*_heterozygote_ = 0.029; OR = 0.645 (0.423–0.984) *p*_codominant1_ = 0.042), non-HDL-C > 160 mg/dL (OR = 0.670 (0.473–0.947) *p*_additive_ = 0.023; OR = 0.571 (0.377–0.865) *p*_dominant_ = 0.008; OR = 0.584 (0.386–0.885) *p*_heterozygote_ = 0.011; OR = 0.558 (0.363–0.856) *p*_codominant1_ = 0.008), and a high risk of LDL pattern B (OR = 2.527 (1.044–6.118) *p*_recessive_ =0.040) and insulin resistance of adipose tissue (OR = 1.624 (1.055–2.501) *p*_dominant_ = 0.028; OR = 1.680 (1.090–2.589) *p*_heterozygote_ = 0.019; OR = 1.705 (1.092–2.664) *p*_codominant1_ = 0.019). On the other hand, rs10412574 was associated with a low risk of the insulin resistance of adipose tissue (OR = 0.522 (0.306–0.890) *p*_recessive_ = 0.017; OR = 0.538 (0.293–0.987) *p*_codominant2_ = 0.045), and a high risk of having aspartate aminotransferase >*p*75 (OR = 1.667 (1.033–2.691) *p*_dominant_ = 0.036; OR = 2.123 (1.145–3.937) *p*_codominant_ =0.017). All the models were adjusted for age, sex, and body mass index (Table 2).

In healthy controls, rs10410870 was associated with a high risk of having low levels of interleukin 10 (<*p*25) (OR = 1.612 (1.062–2.447) *p*_recessive_ = 0.025; OR = 1.590 (1.024–2.467) *p*_codominant2_ = 0.039) and magnesium deficiency (OR = 1.836 (1.053–3.200) *p*_heterozygote_ = 0.032). On the other hand, rs10412574 was associated with a low risk of LDL pattern B (OR = 0.720 (0.562–0.921) *p*_heterozygote_ = 0.009; OR = 0.751 (0.567–0.996) *p*_codominant1_ = 0.047), and a high risk of magnesium deficiency (OR = 2.047 (1.153–3.633) *p*_heterozygote_ = 0.014). All the models were adjusted for age, sex, and body mass index (Table 3).

### 3.3. Association of the rs10410870 Polymorphism with AVC

It has been reported that AVC may be present in individuals without CAC. Previously, we established that AVC is present in 8.5% of subjects without CAC [18]. In the present study, we analyzed 846 controls (with CAC = zero), and 91 of them presented with AVC without CAC. Independent analysis in these healthy controls showed that the rs10410870 polymorphism was associated with a high risk of AVC under six models adjusted for several confounding variables. Under the model adjusted by age, sex, BMI, LDL-cholesterol, Type 2 diabetes mellitus, and current smoking, these individuals presented a 2.2-fold higher risk of AVC (OR = 2.239 (1.209–4.147) *p* = 0.010) (Figure 1).

### 3.4. Haplotype Analysis

The studied polymorphisms were in high linkage disequilibrium (D`x 100 = 95), and only the *TA* haplotype was associated with an increased risk of having high levels of aspartate aminotransferase (≥*p*75) (Table 4).

## 4. Discussion

*FOXA3* is a transcription factor that regulates cholesterol efflux and, in consequence, the atherogenic process. The gene that encodes this factor is polymorphic, and some polymorphisms have been associated with some metabolic parameters [11]. No association studies have been conducted in patients with atherosclerosis. Considering that overexpression of *FOXA3* could reduce atherosclerosis, increasing the reverse cholesterol transport [4], we analyzed the distribution of two *FOXA3* polymorphisms in individuals with and without SA. Our study provides interesting information on the association of the two *FOXA3* polymorphisms with metabolic parameters in two well-defined groups of individuals from a clinical point of view, individuals with AS and healthy subjects. The polymorphisms were not associated with SA; however, rs10410870 was associated with a low risk of having total cholesterol >200 mg/dL and non-HDL-cholesterol >160 mg/dL, and a high risk of having LDL pattern B and insulin resistance of adipose tissue in individuals with SA. This polymorphism was associated with a high risk of having interleukin 10 <*p*25 and magnesium deficiency in controls. On the other hand, rs10412574 was associated with a low risk of having insulin resistance of adipose tissue and a high risk of having aspartate aminotransferase ≥*p*75 in individuals with SA, whereas in controls it was associated with a low risk of having LDL pattern B and a high risk of having a magnesium deficiency. The associations observed in both groups are different, highlighting the association of rs10410870 with a high risk of having insulin resistance adipose tissue in individuals with SA and the association of both polymorphisms with a high risk of magnesium deficiency in the control group. The participation of FOXA3 in adipose tissue has been well established, and it is well-known that the activation of the glucocorticoid receptor upregulates FOXA3 in adipose tissue [7] and promotes adipocyte differentiation through the induction of PPARγ (peroxisome proliferator-activated receptor-gamma) [6]. Ma et al., using Foxa3-null mice fed a high-fat diet, showed that these mice are protected from visceral adipose depot expansion, demonstrating that FOXA3 is an early regulator of adipocyte differentiation [19]. The critical role of the FOXA3 in the cholesterol homeostasis could explain the association of rs10410870 with the low risk of having total cholesterol >200 mg/dL and non-HDL-cholesterol >160 mg/dL, and the high risk of having LDL pattern B that was observed in our study. An effect of FOXA3 cannot explain the association of the polymorphisms with a magnesium deficiency; however, it is known that magnesium deficiency produces an increase in LDL levels. In our study, the rs10410870 polymorphism is associated with a high risk of having LDL pattern B, a pattern characterized by a high proportion of LDL particles that are abnormally small and highly atherogenic. On the other hand, magnesium deficiency has been associated with several clinical conditions, including diabetes, hypertension, insulin resistance, and hyperlipidemia [20,21,22,23]. In our study, 91 individuals presented with AVC without CAC. In this group, we detected an association of the rs10410870 polymorphism with AVC. Follow-up of these individuals is required to see if they will develop CAC. It is important to consider that the results presented in the study are from the transversal phase of the GEA project. At present, we are finishing the prospective phase of the project. In this phase, we will be able to establish if the individuals with AVC developed CAC or not. It has been proposed that the endothelial dysfunction that produces infiltration of inflammatory cells and lipid deposits in the tissues is the mechanism that initiates AVC [24]. In this context, Li et al. [4] demonstrated that FOXA3 regulates the ApoA-I expression, a molecule that may inhibit monocyte activation and, in consequence, inhibit inflammation [25], a process that could be related to the AVC.

Both studied polymorphisms are located in the promoter region. Bioinformatics analysis showed that rs10410870 produces a binding site for the CEBP transcription factor; in this case, the CEBP binds with more affinity to the *A* allele when compared to the *G* allele. CEBP is a family of transcriptional factors involved in the macrophage activation. One of these factors, the CEBP beta, regulates several cytokines such as IL-1B, IL-6, IL-8, IL-12, and TNF alpha, with its consequent role in the inflammatory process [26,27,28,29]. The role of this CEBP in inflammation and atherosclerosis was evaluated by Rahman et al. in a mice model. They detected decreased atherosclerotic lesions in C/EBP β−/− mice compared to irradiated ApoE−/− mice when their bone marrow was reconstituted [28]. On the other hand, CEBP is involved in the lipid metabolism in adipose tissue and the liver [30]. On the other hand, the *T* allele of the rs10412574 polymorphism produces binding sites for HMGIY, HNF4-alpha, SFI, and SRF transcription factors. The HMGIY is a member of the architectural transcription factors of the HGM-1 family with meaningful participation in gene expression and growth regulation [31]. This transcription factor participates in the expression of the chemokine MGSA/GROalpha [32] and endothelial cell adhesion molecule E-selectin [33]. HNF4-alpha is a factor involved in the transcriptional regulation of hepatocyte genes implicated in differentiation, morphogenesis, glucose, and lipid metabolism [34].

Our study has strengths and limitations. Among the strengths, we can highlight having a group of well-characterized individuals from the clinical, demographic, biochemical, and tomography points of view. Our control group only includes those individuals without coronary artery calcium determined by computed tomography (individuals with CAC = zero). Among the limitations, we have the fact that the functional approaches were only defined using bioinformatics tools, and it will be necessary to establish experimental designs that establish the actual functional effect of the polymorphisms included in the study. In addition, we only studied two polymorphisms of the *FOXA3* gene; a study that includes more polymorphisms could help define the proper role of this gene in the genetic susceptibility to AS and cardiovascular risk factors.

Considering that the associations described have not been previously reported, studies in other populations will be necessary to establish whether they are unique to the Mexican population or are replicated in other ethnic groups.

## 5. Conclusions

In summary: our results did not show an association of the *FOXA3* polymorphisms with SA; however, the polymorphisms were associated with some cardiometabolic parameters in individuals with and without SA.

## Figures and Tables

**Figure 1 biomolecules-12-00601-f001:**
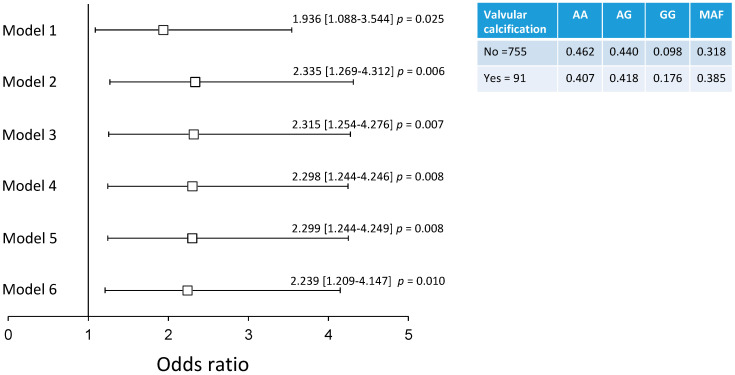
Association of the *FOXA3* rs10410870 genotype with AVC. Model 1: unadjusted; Model 2: adjusted for age and sex; Model 3: Model 2 + adjusted for body mass index; Model 4: Model 3 + adjusted for LDL-cholesterol; Model 5: Model 4 + adjusted for type 2 diabetes mellitus; Model 6: Model 5 + current smoking.

**Table 1 biomolecules-12-00601-t001:** Demographic, clinical, biochemical characteristics, and frequencies of *FOXA3* polymorphisms.

	Total Population	Healthy Controls	SA Individuals	*p*
n	1456	1070	386	
Age (years)	53 ± 9	51 ± 9	59 ± 8	<0.001
Sex (Male%)		41.2	75.6	<0.001
Body mass index (kg/m^2^)	28.0 (25.6–31.0)	27.8 (25.4–30.9)	28.1 (25.9–31.0)	0.060
Total cholesterol (mg/dL)	191 (167–214)	190 (166–211)	198 (169–220)	0.008
LDL-cholesterol (mg/dL)	118 (97–138)	116 (95–134)	124 (102–145)	<0.001
HDL-cholesterol (mg/dL)	44 (36–54)	45 (36–55)	43 (36–50)	0.015
Non-HDL-cholesterol (mg/dL)	144 (122–168)	142 (121–164)	153 (128–175)	<0.001
LDL estimated size	1.21 (1.08–1.38)	1.21 (1.08–1.38)	1.20 (1.09–1.37)	0.972
Magnesium concentration (mg/dL)	2.07 ± 0.17	2.08 ± 0.17	2.05 ± 0.18	0.002
Interleukin-10 concentration (pg/mL)	0.45 (0.24–1.03)	0.45 (0.24–1.03)	0.46 (0.24–1.05)	0.899
Insulin resistance of adipose tissue	9.7 (6.2–14.4)	9.3 (6.1–14.3)	10.4 (6.7–14.5)	0.053
Aspartate amino transferase (UI/L)	25 (21–30)	25 (21–30)	25 (21–30)	0.494
Total cholesterol > 200 mg/dL (%)	39.6	36.7	47.7	<0.001
Non-HDL-cholesterol > 160 mg/dL (%)	32.1	28.5	42.0	<0.001
LDL pattern B (%)	47.2	47.1	47.3	1.000
Insulin resistance of adipose tissue (%)	50.4	47.9	57.7	0.001
Elevated aspartate amino transferase (%)	35.5	36.6	32.4	0.153
Low interleukin-10 concentration (%)	31.2	31.6	30.1	0.641
Magnesium deficiency (%)	5.9	5.2	8.2	0.040
Current smoking (%)	22.5	23.0	21.2	0.523
Aortic valve calcification (%) *	19.5	10.8	43.5	<0.001
*FOXA3* frequency (%)				
rs10412574				
*CC*	32.1	31.9	32.6	0.858
*CT*	47.8	47.7	48.2	
*TT*	20.1	20.5	19.2	
*C*	56.0	55.7	56.7	0.884
*T*	44.0	44.3	43.3	
rs10410870				
*AA*	46.4	46.3	46.9	0.056
*AG*	44.5	43.6	46.9	
*GG*	9.1	10.1	6.2	
*A*	68.7	68.0	70.3	0.512
*G*	31.3	32.0	29.7	

SA: Subclinical atherosclerosis; LDL: Low density lipoprotein; HDL: High density lipoprotein. * Data available in 846 controls. *p* values: AS vs. Controls.

**Table 2 biomolecules-12-00601-t002:** Association of *FOXA3* polymorphisms with metabolic risk factors in SA subject.

	Genotype Frequency			MAF	Model	OR [95% CI]	*p*
Total cholesterol > 200mg/dL							
rs10410870	*AA*	*AG*	*GG*				
No (n = 202)	0.431	0.520	0.050	0.309	Heterozygote	0.633 (0.421–0.954)	0.029
Yes (n = 184)	0.511	0.413	0.076		Codominant 1	0.645 (0.423–0.984)	0.042
Non-HDL-C > 160 mg/dL							
rs10410870	*AA*	*AG*	*GG*				
No (n = 224)	0.415	0.522	0.063	0.324	Additive	0.670 (0.473–0.947)	0.023
Yes (n = 162)	0.543	0.395	0.062	0.259	Dominant	0.571 (0.377–0.865)	0.008
					Heterozygote	0.584 (0.386–0.885)	0.011
					Codominant 1	0.558 (0.363–0.856)	0.008
LDL pattern B							
rs10410870	*AA*	*AG*	*GG*				
No (n = 204)	0.468	0.493	0.039	0.287	Recessive	2.527 (1.044–6.118)	0.040
Yes (n = 182)	0.467	0.445	0.088	0.310			
Insulin resistance adipose tissue							
rs10412574	*CC*	*CT*	*TT*				
No (n = 163)	0.318	0.433	0.248	0.463	Recessive	0.522 (0.306–0.890)	0.017
Yes (n = 223)	0.336	0.500	0.164	0.397	Codominant 2	0.538 (0.293–0.987)	0.045
rs10410870	*AA*	*AG*	*GG*				
No (n = 163)	0.522	0.408	0.070	0.273	Dominant	1.624 (1.055–2.501)	0.028
Yes (n = 223)	0.430	0.514	0.056	0.300	Heterozygote	1.680 (1.090–2.589)	0.019
					Codominant 1	1.705 (1.092–2.664)	0.019
Aspartate aminotransferase ≥ *p*75							
rs10412574	*CC*	*CT*	*TT*				
No (n = 261)	0.360	0.475	0.165	0.402	Dominant	1.667 (1.033–2.691)	0.036
Yes (n = 125)	0.256	0.496	0.248	0.496	Codominant 2	2.123 (1.145–3.937)	0.017

All the models were adjusted for age, sex, body mass index. HDL-C: High density lipoprotein-cholesterol. MAF: Minor allele frequency.

**Table 3 biomolecules-12-00601-t003:** Association of *FOXA3* polymorphisms with metabolic risk factors in healthy controls.

Polymorphism	Genotype Frequency			MAF	Model	OR [95% CI]	*p*
Interleukin 10 < *p*25							
rs10410870	*AA*	*AG*	*GG*				
No (n = 732)	0.464	0.448	0.088	0.311	Recessive	1.612 (1.062–2.447)	0.025
Yes (n = 338)	0.444	0.422	0.134	0.399	Codominant 2	1.590 (1.024–2.467)	0.039
LDL pattern B							
rs10412574	*CC*	*CT*	*TT*				
No (n = 566)	0.304	0.512	0.183	0.440	Heterozygote	0.720 (0.562–0.921)	0.009
Yes (n = 504)	0.335	0.435	0.230	0.447	Codominant 1	0.751 (0.567–0.996)	0.047
Magnesium deficiency							
rs10412574	*CC*	*CT*	*TT*				
No (n = 1014)	0.321	0.468	0.210	0.444	Heterozygote	2.047 (1.153–3.633)	0.014
Yes (n = 56)	0.222	0.648	0.130	0.446			
rs10410870	*AA*	*AG*	*GG*				
No (n = 1014)	0.474	0.422	0.104	0.316	Heterozygote	1.836 (1.053–3.200)	0.032
Yes (n = 56)	0.370	0.574	0.056	0.339			
AVC (%) *							
rs10410870	*AA*	*AG*	*GG*				
No (n = 755)	0.462	0.440	0.098	0.318	Additive	1.464 (1.052–2.038)	0.024
Yes (n = 91)	0.407	0.418	0.176	0.385	Recessive	2.315 (1.254–4.276)	0.007
					Codominant 2	2.481 (1.277–4.819)	0.007

All the models were adjusted for age, sex, body mass index. MAF: Minor allele frequency; AVC: Aortic Valve Calcification. * Data were available in 846 contol subjects.

**Table 4 biomolecules-12-00601-t004:** Association of *FOXA3* haplotypes with SA and with cardiovascular risk factors in SA and healthy controls.

Haplotypes		Subclinical Atherosclerosis		OR [95% CI]	*p*
		**Yes**	**No**		
H1	*TA*	0.427	0.436	0.964 (0.817–1.139)	0.673
H2	*CG*	0.291	0.312	0.905 (0.755–1.083)	0.278
H3	*CA*	0.276	0.245	1.175 (0.976–1.416)	0.090
		**Coronary Risk Factors**			
Subclinical Atherosclerosis					
		Total cholesterol > 200 mg/dL			
		Yes	No		
H1	*TA*	0.447	0.409	1.193 (0.896–1.588)	0.226
H2	*CG*	0.279	0.303	0.895 (0.655–1.221)	0.485
H3	*CA*	0.270	0.281	0.945 (0.689–1.296)	0.728
		Cholesterol non-HDL > 160 mg/dL			
		Yes	No		
H1	*TA*	0.462	0.402	1.276 (0.956–1.703)	0.097
H2	*CG*	0.255	0.318	0.7391 (0.537–1.016)	0.063
H3	*CA*	0.279	0.274	1.0276 (0.747–1.413)	0.867
		LDL pattern B			
		Yes	No		
H1	*TA*	0.419	0.434	0.937 (0.703–1.247)	0.655
H2	*CG*	0.306	0.279	1.137 (0.833–1.553)	0.416
H3	*CA*	0.271	0.280	0.956 (0.697–1.313)	0.785
		Insulin resistance of adipose tissue			
		Yes	No		
H1	*TA*	0.405	0.464	0.780 (0.581–1.075)	0.098
H2	*CG*	0.304	0.273	1.156 (0.837–1.596)	0.376
H3	*CA*	0.282	0.262	1.115 (0.803–1.548)	0.515
		Aspartate aminotransferase ≥ *p*75			
		Yes	No		
H1	*TA*	0.486	0.399	1.438 (1.062–1.949)	0.018
H2	*CG*	0.250	0.311	0.748 (0.532–1.052)	0.095
H3	*CA*	0.254	0.287	0.835 (0.593–1.177)	0.304
Controls					
		Aortic Valve Calcification			
		Yes	No		
H1	*TA*	0.366	0.438	0.746 (0.543–1.025)	0.071
H2	*CG*	0.372	0.310	1.328 (0.964–1.828)	0.081
H3	*CA*	0.249	0.244	1.019 (0.713–1.456)	0.916
		IL-10 < *p*25			
		Yes	No		
H1	*TA*	0.414	0.445	0.833 [0.730–1.064)	0.198
H2	*CG*	0.339	0.306	1.162 [0.952–1.419)	0.139
H3	*CA*	0.240	0.244	1.486 [0.845–1.300)	0.667
		LDL pattern B			
		Yes	No		
H1	*TA*	0.444	0.430	1.060 (0.893–1.258)	0.504
H2	*CG*	0.308	0.316	0.965 (0.803–1.160)	0.711
H3	*CA*	0.245	0.245	0.999 (0.819–1.218)	0.993
		Magnesium deficiency			
		Yes	No		
H1	*TA*	0.451	0.438	1.065 (0.722–1.572)	0.749
H2	*CG*	0.340	0.308	1.170 (0.777–1.760)	0.451
H3	*CA*	0.206	0.247	0.777 (0.482–1.257)	0.306

OR: Odds ratio; CI: Confidence intervals.

## Data Availability

The data presented in this study are available upon request from the corresponding author.

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
