# Peer review of "FOXA3* Polymorphisms Are Associated with Metabolic Parameters in Individuals with Subclinical Atherosclerosis and Healthy Controls—The GEA Mexican Study"

_biomolecules, 2022, doi:10.3390/biom12050601_

Round 1
Reviewer 1 Report
This is an original research article. Authors aimed to examine potential association between two FOXA3 polymorphisms (rs10410870 and rs10412574) and subclinical Atherosclerosis (SA), using data from the GEA Mexican study including 386 individuals with SA and 1070 controls. Using several statistical methods including logistic regression analysis under different models of inheritance, they tested potential association adjusted to different confounding variables. No association with SA was identified. However, several associations with some metabolic trait were identified in both SA and control groups. Some these association were different based on the genotype. Authors conclude no association between SA and FOXA3 polymorphisms. However, independent analysis 846 healthy controls revealed association between rs10410870 and the risk for vascular calcification. Major concerns need to be addressed before considering potential publication:
- Limited Novelty and Impact. The role of FOXA3 polymorphisms in metabolism and adiposity have been established. Specifically, the reported an association of the rs28666870 polymorphisms with greater total lean body mass, increased BMI, and appendicular lean mass. However, no association studies of FOXA3 polymorphisms with atherosclerosis or metabolic parameters have been done. Investigating the association between FOXA3 and SA could potentially lead to incremental knowledge of potential FOXA3 role in SA function. However, the results did not reveal any association with SA. Further, several associations were found in both SA and control groups. Therefore, the novelty and impact are limited.
- Sample size for SA group is small.
- The study is limited to association analysis, but did not address potential function. Several functional genomics methods were proposed in the methods section (splicing port, SNP function, etc). However, the authors did not present the results of these analysis, nor they mentioned any potential prediction of the functional prediction of these polymorphisms.
- Per authors definition, the healthy controls ar individuals with 77 CAC score = 0. However, they performed independent analysis 846 healthy controls and revealed association between rs10410870 and the risk for vascular calcification. This result is confusing as it is not clear how 91 healthy control induvial in this cohort had vascular calcification.
- The studied polymorphisms were in high linkage disequilibrium (D`x 100 = 95); however, none of the haplotypes was associated with SA and cardiometabolic parameters (data not 178 shown).
These data need to be presented in this manuscript. - Overall, this is a descriptive study and revealed negative association with the primary outcome (SA). The discussion section is speculative. No attempt to perform mechanistic studies in cells or model organisms.
Author Response
1.- Limited Novelty and Impact. The role of FOXA3 polymorphisms in metabolism and adiposity has been established. Specifically, they reported an association of the rs28666870 polymorphisms with greater total lean body mass, increased BMI, and appendicular lean mass. However, no association studies of FOXA3 polymorphisms with atherosclerosis or metabolic parameters have been done. Investigating the association between FOXA3 and SA could potentially lead to incremental knowledge of the potential FOXA3 role in SA function. However, the results did not reveal any association with SA. Further, several associations were found in both SA and control groups. Therefore, the novelty and impact are limited.
Answer: As commented by the reviewer there is only one paper that analyzed the association of the FOXA3 polymorphisms with metabolic parameters published in 2015 and there are no studies in SA. This study included 392 children, adolescents, and young adults. In this work, the rs28666870 polymorphism was associated with greater total lean body mass, increased BMI, and appendicular lean mass (this manuscript is cited in our work). After that, no studies of the FOXA3 polymorphisms in metabolic or cardiovascular diseases have been reported.
We added the phrase “Given the role that FOXA3 plays in macrophage cholesterol efflux and its possible association with insulin resistance, obesity, and atherosclerosis, we consider it interesting to carry out studies on the association of polymorphisms of this gene with cardiovascular diseases and metabolic parameters.” in the introduction section.
The phrase “Our study provides interesting information on the association of two FOXA3 polymorphisms with metabolic parameters in two well-defined groups of individuals from a clinical point of view, individuals with AS and healthy subjects.” has been added in the discussion section.
2.- The sample size for the SA group is small.
Answer: As commented by the reviewer the sample size for SA seems small. In order to define if this number of individuals is enough to detect an association between the studied polymorphisms and SA, we determined the statistical power using the OpenEpi software (http://www.openepi.com/Power/PowerCC.htm)2 . With this software, we established that the statistical power to detect an association between the FOXA3 polymorphisms and SA was 0.80.
The phrase “The statistical power to detect association of the polymorphisms with SA was determined using the OpenEpi software [http://www.openepi.com/Power/PowerCC.htm (accessed on march 30)]” was added in the statistical analysis section.
The phrase “The statistical power to detect an association with SA considering an unmatched case-control study was 0.80” was added in the results section.
3.- The study is limited to association analysis but did not address potential function. Several functional genomics methods were proposed in the methods section (splicing port, SNP function, etc). However, the authors did not present the results of this analysis, nor did they mention any potential prediction of the functional prediction of these polymorphisms.
Answer: As is commented by the reviewer, we used some bioinformatics tools in order to define the possible functional effect of the studied polymorphisms. Also, we performed an analysis of the genome-wide expression quantitative trait loci (eQTL) dataset. Results of this analysis are included in the discussion section in a paragraph:
“Both studied polymorphisms are located in the promoter region. Bioinformatics analysis showed that the rs10410870 produces a binding site for the CEBP transcription factor; in this case, the CEBP is binding with more affinity to the A allele when compared to the G allele. CEBP is a family of transcriptional factors involved in the macrophage activation. One of these factors, the CEBP beta, regulates several cytokines such as IL-1B, IL-6, IL-8, IL-12, and TNF alpha with its consequent role in the inflammatory process [26-29]. The role of this CEBP in inflammation and atherosclerosis was evaluated by Rahman et al., in a mice model. They detected decreased atherosclerotic lesions in C/EBP β−/− mice to irradiated ApoE−/− mice when its bone marrow was reconstituted [28]. On the other hand, CEBP is involved in the lipid metabolism in adipose tissue and the liver [30]. On the other hand, the T allele of the rs10412574 polymorphism produces binding sites for HMGIY, HNF4-alpha, SFI, and SRF transcription factors. The HMGIY is a member of the architectural transcription factors of the HGM-1 family with meaningful participation in gene expression and growth regulation [31]. This transcription factor participates in the expression of the chemokine MGSA/GROalpha [32] and endothelial cell adhesion molecule E-selectin [33]. HNF4-alpha is a factor involved in the transcriptional regulation of hepatocyte genes implicated in differentiation, morphogenesis, glucose, and lipid metabolism [34].”
Also, the phrase “These tools showed that both polymorphisms have possible functional effects. The rs10410870 produces a binding site for the CEBP transcription factor, whereas the rs10412574 produces binding sites for HMGIY, HNF4-alpha, SFI, and SRF transcription factors.” has been added in section 2.4. Functional analysis.
4.- Per the authors definition, the healthy controls are individuals with CAC score = 0. However, they performed independent analysis 846 healthy controls and revealed association between rs10410870 and the risk for vascular calcification. This result is confusing as it is not clear how healthy control individuals in this cohort had vascular calcification.
Answer: The association that we detected was with valvular calcification. In order to clarify this point, the phrase “It has been reported that AVC may be present in individuals without CAC. Previously we established that AVC is present in 8.5% of subjects without CAC [18]. In the present study, we analyzed 846 controls (with CAC = zero), and 91 of them presented AVC.” has been added in the results section (3.3. Association of the rs10410870 polymorphism with valvular calcification).
5.- The studied polymorphisms were in high linkage disequilibrium (D`x 100 = 95); however, none of the haplotypes was associated with SA and cardiometabolic parameters (data not shown). These data need to be presented in this manuscript.
Answer: As is suggested by the reviewer, the results of the haplotypes have been included as table 4.
6.- Overall, this is a descriptive study and revealed negative association with the primary outcome (SA). The discussion section is speculative. No attempt to perform mechanistic studies in cells or model organisms.
Answer: We agree with the reviewer; our work is an association genetic study. Using bioinformatics tools, we define the possible function effect of the polymorphisms included in the study; however, no experimental designs were done. This last is considered a limitation of our study and the phrase “Among the limitations, we have the fact that the functional approaches were only defined using bioinformatics tools, and it will be necessary to establish experimental designs that establish the actual functional effect of the polymorphisms included in the study.” is included in the discussion section.
Reviewer 2 Report
The article is of clinical and research interest.
Minor comments:
1) If the data are part of some other study (Genetics of Atherosclerotic Disease (GEA) Mexican Cohort), it is recommended that reference be made to the publication of the primary study.
2) In Table 1, it is recommended to specify the ratio of male or female in the sex line.
3) Association of the rs10410870 polymorphism with valvular calcification was studied in 846 volunteers. Table 1 does not include this category. It is recommended to add information about these people and how they were selected.
4) It is recommended to add information on the studied FOXA3 gene polymorphism (rs10410870 and rs10412574) from other studies.
Author Response
1.- If the data are part of some other study (Genetics of Atherosclerotic Disease (GEA) Mexican Cohort), it is recommended that reference be made to the publication of the primary study.
Answer: The results of this cohort have been published in more than 70 papers, some of which are cited in the present paper. Unfortunately, we did not publish the description of the transversal phase of the cohort. Now we are working on the prospective study of the cohort, and we are preparing the manuscript with the description of this phase of the study.
2) In Table 1, it is recommended to specify the ratio of male or female in the sex line.
Answer: This point has been clarified:
Sex (Male %) has been added in table 1.
3) Association of the rs10410870 polymorphism with valvular calcification was studied in 846 volunteers. Table 1 does not include this category. It is recommended to add information about these people and how they were selected.
Answer: The percentage of individuals with aortic valve calcification was included in table 1. The aortic valve calcification was present in 11.2% of the healthy controls and in 43.5 of the individuals with SA. The aortic valve calcification was defined as the same that the CAC using the Agatston method. The phrase “The CAC and AVC were defined using the Agatston method [14].” was added in the material and methods section (2.1. Subjects)
4) It is recommended to add information on the studied FOXA3 gene polymorphism (rs10410870 and rs10412574) from other studies.
Answer: To our knowledge, the only study that includes the analysis of one of these polymorphisms is that of Adler-Wailes et al., which reported an association of the rs28666870 polymorphism with greater total lean body mass, increased BMI, and appendicular lean mass. This manuscript is cited and discussed in our work.
Reviewer 3 Report
The article by Dr G Vargas-Alarcón et al, entitled "FOXA3 polymorphisms are associated with metabolic parame-ters in individuals with subclinical atherosclerosis and healthy controls. The GEA Mexican Study” analyzes the transcription factor FOXA3 and its role in subclinical atherosclerosis (SA) and in the regulation of cardiometabolic parameters, specifically by studying its polymorphisms. Examination of a control group and of AS patients identified two polymorphisms (rs10410870 and rs10412574) that appear to be differentially expressed in the two groups. The study concludes by suggesting that the two polymorphisms are not associated with AS, but with cardiometabolic changes in both groups.
Although the study did not achieve the desired results because a correlation between SA and the FOXA3 polymorphisms was not found, I believe that the basic research and work organization are adequate. I also appreciate the authors' willingness to take up the work initiated by the Adler-Wailes group.
The topic is introduced in an appropriate manner with a reasonable number of correlating references.
The manuscript is generally well organized and fluidly written, but there are still some aspects to improve before it can be published.
- I would suggest including a table with the main characteristics listed in section 2.1 to make the data of the recruited patients more immediate;
- I would suggest indicating whether the two polymorphisms are present in all 1456 individuals enrolled in the study or whether screening analyzes were performed first. Most importantly, I would indicate in how many subjects the rs10410870 variant is present and in how many the rs10412574 variant is present.
- I would suggest expanding section 2.4 "Functional Analysis" rather than just providing a list of links.
- In paragraph 3.4 "Haplotype analysis", I would suggest at least introducing or schematizing the data not shown, as this could strengthen the work.
Author Response
1.- I would suggest including a table with the main characteristics listed in section 2.1 to make the data of the recruited patients more immediate.
Answer: As suggested by the reviewer, in table 1 a column with data from the whole studied population is included.
2.- I would suggest indicating whether the two polymorphisms are present in all 1456 individuals enrolled in the study or whether screening analyzes were performed first. Most importantly, I would indicate in how many subjects the rs10410870 variant is present and in how many the rs10412574 variant is present.
Answer: The screening analysis was not performed; the polymorphisms were determined in all 1456 subjects. Table 1 included the allele and genotype frequencies of each polymorphism to show how many individuals presented the variant for each polymorphism in the whole population, healthy controls, and SA.
3.- I would suggest expanding section 2.4 "Functional Analysis" rather than just providing a list of links.
Answer: As suggested by the reviewer, information on the possible functional effects of the polymorphisms is included in section 2.4. Functional Analysis, the phrase “These tools showed that both polymorphisms have possible functional effects. The rs10410870 produces a binding site for the CEBP transcription factor, whereas the rs10412574 produces binding sites for HMGIY, HNF4-alpha, SFI, and SRF transcription factors.” has been added.
4.- In paragraph 3.4 "Haplotype analysis", I would suggest at least introducing or schematizing the data not shown, as this could strengthen the work.
Answer: To show complete data, we included a table with the haplotypes' association with SA and cardiovascular risk factors (Table 4).
Round 2
Reviewer 1 Report
Authores have made appropriate revisions and addressed the majority of comments in a scientific sound manner. These revisions have imporved the clarity and the merit of the article.
One items [#4] remains for further clarification:
Authors response: [It has been reported that AVC may be present in individuals without CAC. Previously we established that AVC is present in 8.5% of subjects without CAC [18]. In the present study, we analyzed 846 controls (with CAC = zero), and 91 of them presented AVC.”]
- I suggest revising the last sentence:
[and 91 of them presented AVC without CAC]
- These 91 individual may require further longtudinal follow up for potential development of CAC. Please discusse the plan for follow up in the discussion section.
Author Response
1.- I suggest revising the last sentence:
[and 91 of them presented AVC without CAC]
Answer: The sentence has been modified, and the term “without CAC” has been included
New phrase: It has been reported that AVC may be present in individuals without CAC. Previously we established that AVC is present in 8.5% of subjects without CAC [18]. In the present study, we analyzed 846 controls (with CAC = zero), and 91 of them presented AVC without CAC.
2.- These 91 individual may require further longtudinal follow up for potential development of CAC. Please discusse the plan for follow up in the discussion section.
Answer: The GEA project is a cohort with two phases, one transversal, and the other prospective. The results reported in this manuscript were obtained in the transversal phase. Recently we finished the prospective phase, and now we are analyzing the results. With this analysis, we will be able to establish if the individuals with AVC developed CAC or not. This plan is discussed, and the phrase “In our study, 91 individuals presented AVC without CAC. In this group, we detected an association of the rs10410870 polymorphisms with AVC. Follow-up of these individuals is required to see if they will develop CAC. It is important to consider that the results presented in the study are from the transversal phase of the GEA project. At present, we are finishing the prospective phase of the project. In this phase, we will be able to establish if the individuals with AVC developed CAC or not..” has been included in the discussion section.